



# Aero-elastic loads on a 10 MW turbine exposed to extreme events selected from a year-long Large-Eddy Simulation over the North Sea

Gerard Schepers[1], Pim van Dorp[2], Remco Verzijlbergh [2]. Peter Baas[2], Harmen Jonker [2]

[1]ECN Part of TNO, Wind Energy, Petten, 1755LE, The Netherlands
[2]Whiffle, Delft, 2629JD, the Netherlands

*Correspondence to*: J.G Schepers (Gerard.schepers@tno.nl)

**Abstract.** In this article the aero-elastic loads on a 10 MW turbine in response to unconventional wind conditions selected from a year long Large Eddy Simulation on a site at the North Sea are evaluated. Thereto an assessment is made of the practical

importance of these wind conditions within an aero-elastic context based on high fidelity wind modelling. Moreover the accuracy of BEM based methods for modelling such wind conditions is assessed. The study is carried out in a joint effort by the Energy Research Centre of the Netherlands ECN part of TNO and the Dutch meteorological consultancy company Whiffle.

## 1 Introduction

This article describes a study in which the loads are assessed as calculated on a 10 MW wind turbine in response to extreme

wind events on the North Sea. The turbine on which the loads are calculated is the 10 MW Reference Wind Turbine as designed in the EU project AVATAR (Sieros, et al., January 2015).

The study is carried out within the Dutch national project DOWA in a cooperation between ECN part of TNO and Whiffle.

The extreme wind events have been selected from a year-long simulation with the operational LES code GRASP from Whiffle (Gilbert, et al., 2019). GRASP is an atmospheric LES model nested in a global weather model which allows the detailed

modelling of meteorological phenomena on a spatial and temporal grid resolution which is fine enough for aero-elastic load calculations. The resulting extreme wind events are then fed as wind input to the aero-elastic solver PHATAS from WMC (now LM) as used by ECN part of TNO (Lindenburg, 2005). PHATAS is coupled to the AeroModule which is a code with two aerodynamic models, a Blade Element Momentum (BEM) method and a Free Vortex Wake (FVW) method AWSM (Boorsma, Grasso, & Holierhoek, 2012). The calculated loads as response to these extreme wind events are compared with the

loads from a reference design load spectrum which is available from the AVATAR project (Stettner, et al., April 2015) This reference design load spectrum is calculated with a conventional procedure along the IEC standards. By comparing the loads in response to the extreme events with those from the conventional design load spectrum, the importance of extreme wind events can be assessed for practical (load) purposes.

The structure of the present article is as follows:

•  In section 2 the goal of the study is explained.





- In section 3 a short description is given of the turbine on which the load calculations are performed. It also describes the location where the turbine is located.
- Section 4 describes the load modelling of extreme wind events. It explains the GRASP model and the selected extreme events with the validation using measurements. It also describes the rotor modelling from PHATAS and AeroModule and the interface between PHATAS and GRASP.
- Section 5 describes the calculation of the reference design load spectrum.
- In section 6 the comparison between the loads from the extreme events and those from the reference spectrum is given together with an evaluation of results. Special attention will be given to the analysis of results at an extreme Low Level Jet, since these events are often believed to have significant impact on turbine loading, see e.g. (Duncan, November 2018)
- Finally, in section 7 the conclusions and recommendations are given.

## 2 Goal

The goal of the present study is defined as:

- To demonstrate the coupling between the LES code GRASP and the aero-elastic code PHATAS; GRASP delivers the wind input to PHATAS as an alternative to the default wind modelling commonly used in aero-elastic load modelling which is based on stochastic wind field methods
- To assess the impact of extreme wind events on the load spectrum of a representative 10 MW turbine. These extreme events are selected from a year-long GRASP simulations. The loads are compared with those from a standard design load spectrum as calculated by PHATAS in a conventional way along IEC standards. In this way it can be assessed whether these extreme events yield loads which exceed the design load spectrum. If true, this might lead to a recommendation on adaptation of the load spectrum with events found in the present study.
- To assess the accuracy of a standard BEM model for the calculation of the selected extreme wind events. Thereto the loads as calculated by a BEM model are compared with those from a higher fidelity code free vortex wake method AWSM.

## 3 Reference turbine and location

The turbine on which the investigations are carried out is the AVATAR Reference Wind Turbine (RWT) (Sieros, et al., January 2015). This is a turbine with a rated power of 10 MW as designed in the EU project AVATAR. This project was carried out from 2013 until the end of 2017. AVATAR was coordinated by ECN(TNO) and it had a consortium of 13 partners, including GE and LM as industrial partners.

The AVATAR RWT is a low induction variant of a 10 MW RWT which was designed in another EU project, INNWIND.EU (Bak, et al., 2013). The INNWIND.EU RWT has a diameter of 178 meter. The low induction concept used in the AVATAR RWT makes an increase in rotor diameter possible to D= 205.8 meter with a limited increase in loads. The hub height of the AVATAR RWT is 132.7 meter by which the lowest point of the rotor plane is at an altitude of 29,8 meter and the upper part of the rotor plane is at 235,6 meter.

The rated rotor speed is 9.8 rpm which leads to a tip speed of 103.4 m/s;

All design data (the aerodynamic and aero-elastic data of blades, tower, shaft and other components) of the AVATAR RWT are publicly available (Sieros, et al., January 2015). Moreover the design load spectrum has been calculated (Stettner, et al.,



April 2015). This design load spectrum will be used as reference to the loads as calculated in response to the extreme events from GRASP. The calculations of the design load spectrum have been repeated with the most recent versions of design tools

to assure consistency in tools, see section 5.

The site where the turbine is placed is the location of the Meteorological IJmuiden (MMIJ) in the North Sea, 85 km offshore from the Dutch shore (N52°50.89' E3°26.14'). Hence, it is this location where the wind input is calculated by the Whiffle code GRASP.

**4 Calculation of loads for extreme events**

**4.1 Wind input**

**4.1.1 Grasp**

The GRASP code is a Large Eddy Simulation (LES) model developed by Whiffle that is based on the Dutch Atmospheric Large Eddy Simulation (DALES). The LES code runs on Graphics Processing Units (GPUs) and is therefore referred to as

GRASP: GPU-Resident Atmospheric Simulation Platform. GRASP can be run with boundary conditions from a large scale-weather model (Gilbert, et al., 2019) For this study, GRASP has been run for the location of the Meteo Mast IJmuiden in the Dutch North Sea area with ERA5 boundary conditions. Driving the LES with boundary conditions from a large-scale weather model, ensures that the full spectrum of atmospheric flow from synoptic to turbulent scales is considered. Amongst others, the interaction between atmospheric stability, turbulence and shear is resolved.

A full year of LES runs of 24 hours each has been performed on a resolution of 20m. From these results, several extreme wind events have been identified, including Low Level Jets, high shear, high veer, strong gusts, fast ramps and high turbulence cases. These cases have been re-run and used as boundary conditions for a higher resolution run in the concurrent precursor setting. Thereto a three-way nested simulation has been carried out, see Fig.1 at 8, 4, and 2 meter resolution with 256 grid boxes in each direction which gives a domain size of 2x2 km$^2$, 1x1 km$^2$ and 500x500 m$^2$ respectively. The finest grid with a

resolution of 2 meter yields 51 wind speed points over the 103 meter AVATAR blade radius. The finest temporary resolution is 10 Hz which yields an azimuth interval of 6 degrees at the rated rotor speed of 10 rpm.



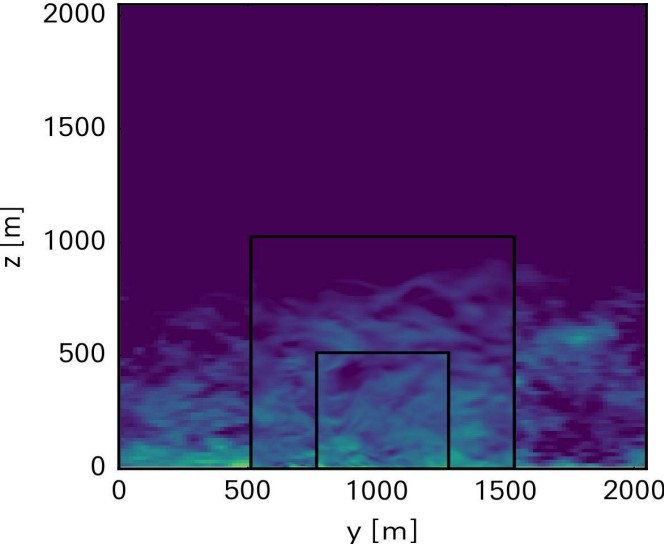

Figure 1 : Vertical cross-sections of wind speed in the three different nested LES runs. The coarsest runs uses periodic lateral boundary conditions and large-scale forcing from ERA5. The higher resolution runs use lateral boundary conditions from the 'upper' nests.

### 4.1.2 Selection of extreme events

The period for which the GRASP simulations were carried out was from 2014/12/1 to 2015/12/1. The calculational domain of the nested simulations was centered around the hub height of 133 meters with a spatial resolution of 2 meters and a temporary resolution of 0.1 seconds as mentioned in section 4.1.1. The considered wind speeds are between 5 and 25 m/s i.e. between the cut-in and cut-out wind speed of the AVATAR RWT

Then the following five "extreme" cases of 10 minutes were selected.

- Strongest low-level jet (LLJ). Note that LLJ's were detected with the algorithm from (Baas, 2009)
- Strongest wind veer over the rotor
- Strongest shear over the rotor
- Highest turbulence kinetic energy (TKE) below cut-out wind speed
- Highest turbulence intensity (TI) around rated wind speed (i.e. higher than 10 m/s) and lower than cut-out

In the Figs 2 to 4, the 10-minute averaged profiles of the extreme low level jet, veer and shear are plotted in terms of wind speed and wind direction as function of height. The figures also indicate the lowest and upper part of the rotor plane as well as the hub height.

For the strongest low level jet, it can be seen that the wind speed at the lowest point of the rotor plane is approximately 9.2 m/s. Going upward it increases to a maximum value of almost 13 m/s. This value is reached slightly below hub height. Then above hub height the wind speed decreases to approximately 10.3 m/s at the upper part of the rotor plane. The wind speed



variation with height goes together with a relatively large veer from approximately 230 degrees at the lowest point of the rotor plane to 239 degrees slightly below hub height above which it remains more or less constant. It must be noted that a shear exponent of 0.2 (i.e. the exponent used in the IEC reference load spectrum, see section 5) at a comparable hub height wind

speed of 13 m/s yields a velocity of 9.7 m/s at the lower part of the rotor plane. In other words, the shear prescribed by the standards is only slightly less than the shear from the LLJ in the lower part of the rotor plane.

The strongest wind veer case shows a wind direction of approximately 85 degrees at the lowest part of the rotor plane and a wind direction of approximately 120 degrees at the upper part, leading to a wind direction difference of 35 degrees.

The strongest shear case shows a wind speed of approximately 11.5 m/s at the lowest part of the rotor plane above which it

increases to almost 16 m/s at hub height above which it increases further to approximately 19 m/s at the upper position of the rotor plane. Although a wind speed difference of 8.5 m/s over the rotor plane is seemingly large it must be noted that a wind shear exponent of 0.2 (i.e. the exponent prescribed in the standards for the normal operating condition cases) and a hub height wind speed of 16 m/s already gives a wind speed difference of 6.2 m/s over the rotor plane.

For the case with extreme turbulence intensity and extreme turbulent kinetic energy the turbulence intensities at hub height

are found to be approximately 5% and 6.5% at approximately 14.8 m/s and 22.5 m/s respectively. It is noted that although these turbulence intensities are the highest for the selected year, they are much lower than the values for turbulence class A at the corresponding wind speeds (approximately 18% and 16%). This indicates that the reference design load spectrum as calculated in the AVATAR project is conservative for isolated turbines at the selected site. However even a turbulence class C (the lowest possible turbulence class in IEC) leads to turbulence intensities which are still far below the extreme

turbulence intensities in the selected year.

It is also important to note that the extreme shear and extreme low level jet cases go together with very low turbulence levels. This is shown in table 1, which gives the turbulence intensity as function of height for the LLJ event.

| Height [m] | Turbulence intensity [%] |
|---|---|
| 31 | 5.8 |
| 81 | 3.3 |
| 133 | 1.6 |
| 185 | 1.3 |
| 235 | 1.2 |

Table 1 Turbulence intensity as function of height for the extreme Low Level Jet case

The turbulence intensity at hub height is 1.6%. This low turbulence intensity should be kept in mind when analyzing the load results. The turbulence intensity decreases from 1.6% at hub height to 1.2% at h = 235 meter despite the decreasing wind

speed above hub height in Fig. 2. This implies that the decreasing turbulence intensity with height should be attributed to a strong decrease in standard deviation of wind speed fluctuations which overcompensates the decreasing wind speed. Under 'normal' atmospheric conditions the decreasing turbulence intensity with height would largely be a result of the increasing wind speed with altitude.

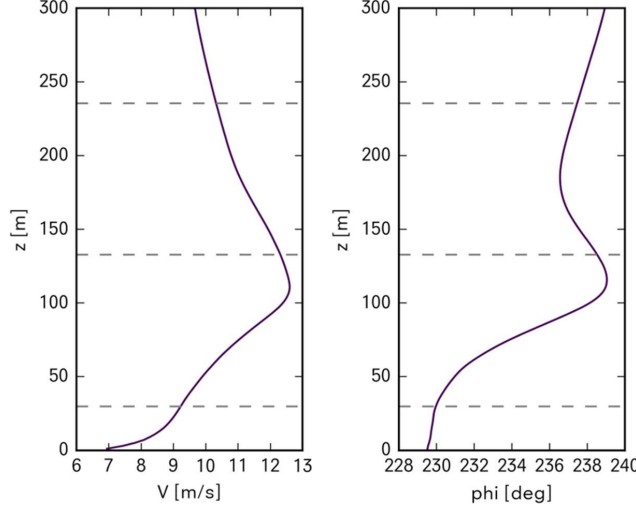

Figure 2: Extreme low level jet event: Wind speed and wind direction as function of height, also indicated are hub height and lower/upper
height of rotor plane



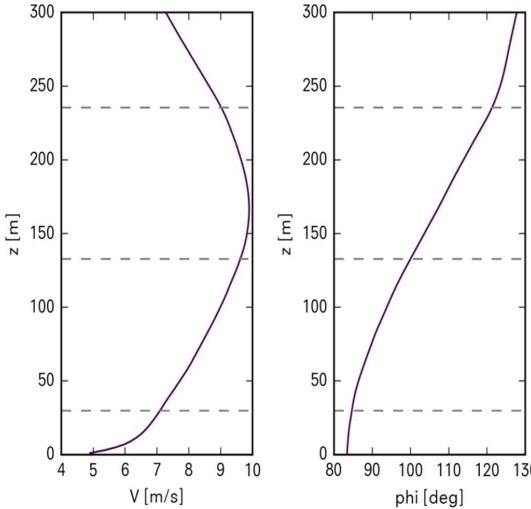

Figure 3: Extreme veer event: Wind speed and wind direction as function of height, also indicated are hub height and lower/upper height of rotor plane

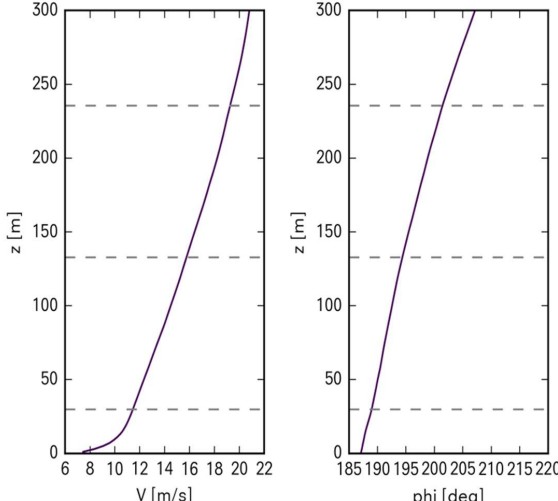

Figure 4: Extreme shear event: Wind speed and wind direction as function of height, also indicated are hub height and lower/upper height of rotor plane





### 4.1.3 Validation of extreme events

As mentioned in section 3 the GRASP wind simulations were done for the location of Meteorological Mast IJmuiden (MMIJ).
This offers a way to validate the GRASP calculations with the measurements from that mast.

The mast is shown in Fig. 5 and the instrumentation of the mast is given in (Werkhoven & Verhoef, 2012). Measurements are taken with anemometers on a mast which are placed at three different heights above sea level, i.e.: 27m, 58m and at the top level of 92 meter (note that some wind speed sensors are mounted at an altitude of 85 meters as well).
They are combined with LIDAR measurements which are taken at 90, 115, 140, 165, 190, 215, 240, 265, 290, 315 meter above sea level.

For each of the selected cases vertical profiles of the relevant variables are given from 2 hours before until 2 hours after the occurrence of the extreme event (indicated as -2h and 2h respectively, with t=0 the moment where the extreme events from section 4.1.2 are calculated). The area of the rotor blade is again indicated with horizontal dashed lines. Results are shown
from the various LES simulations with different resolutions where also the ERA5 reanalysis results are shown. The simulations with finer resolution were only run for a few hours and until t=0 only. This explains why not all simulations are presented each hour.

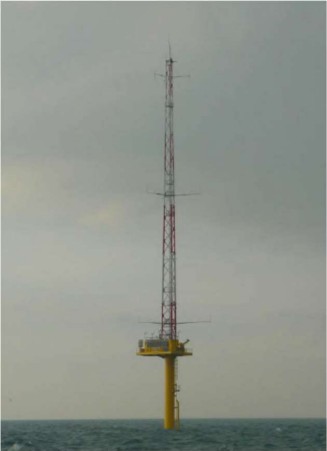

Figure 5: Meteorological Mast IJmuiden

**Low-level jet (LLJ)**

Figure 6 shows the LLJ case. The measurements do not show a typical LLJ profile, although especially at t=2h a wind maximum is present. GRASP represents the wind speed profile much better than ERA5. There is little effect of a different resolution.





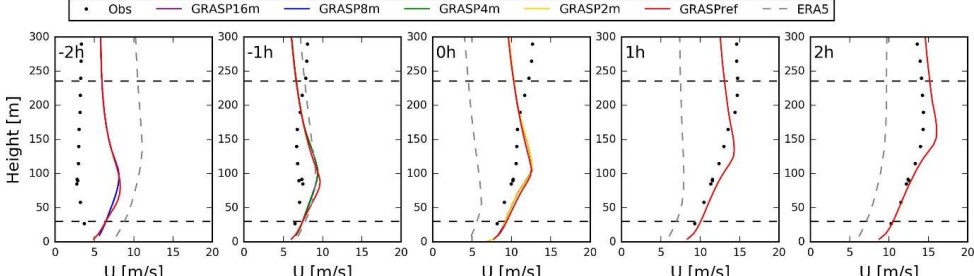

Figure 6. Profiles of wind speed for the low-level jet case.

In the sequel of this report emphasis is put on the loads at Low Level Jets and because the agreement for this particular case is slightly poorer than expected some further analysis was carried out by comparing more measured and calculated LLJ profiles in the simulated year to investigate whether a better agreement is found at other moments. Generally speaking this is the case indeed.

    A representative example is given in Fig. 7. This figure shows a sequence of two subsequent 10-minute averaged wind speed

profiles, measured and calculated taken at exactly the same time (May 11, 2015 at 13hrs 40 and 13 hrs 50). An (Excel polynomial) curve fitting was applied on the calculated and measured wind speed profiles.

    Despite some differences in level and shear, Fig. 7 generally shows a good agreement between calculated and measured LLJ. It must be noted that an indication on measurements uncertainty can be found by comparing the LIDAR and mast measurements which at an altitude of approximately 90 meter are taken at almost the same height. As such both measurements

should indicate the same value but this is not exactly the case. Mast measurements are a well proven technology and as such it may be expected that LIDAR measurements have the largest uncertainty which may become poorer with increasing altitudes and low turbulence intensity (due to the fact that the number of aerosols decrease with altitude and decreasing turbulence intensity). It is then recalled from table 1 that LLJ's events go together with very low turbulence intensities by which the uncertainty for the LIDAR measurements at these events may be larger.

**Wind shear**

Figure 8 shows a strong wind shear over the rotor plane in both measurements and calculations.

**Wind Veer**

Figure 9 shows that a large variation of wind direction with height is found in both calculations as measurements. ERA5 underestimates the turning of the wind with height.


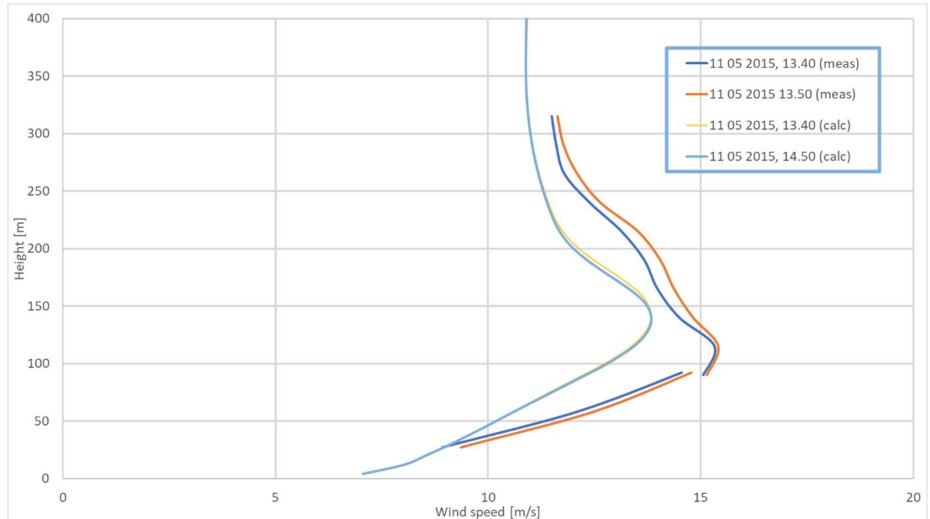

Figure 7. Comparison between measured and calculated LLJ profiles for 2 subsequent 10-minute averaged profiles .

**205**     **Turbulent Kinetic Energy (TKE)**

Figure 10 shows profiles of TKE. The GRASP run with a grid-spacing of 2m is very close to the measurements. Coarsening the resolution leads to higher values of TKE. Note that no ERA5 results are shown, as this model does not resolve turbulent motions.

**Turbulence Intensity (TI)**

**210**     Figure 11 shows profiles of TI. As for the TKE case, the simulation with 2-m grid-spacing reproduces the observations very well.



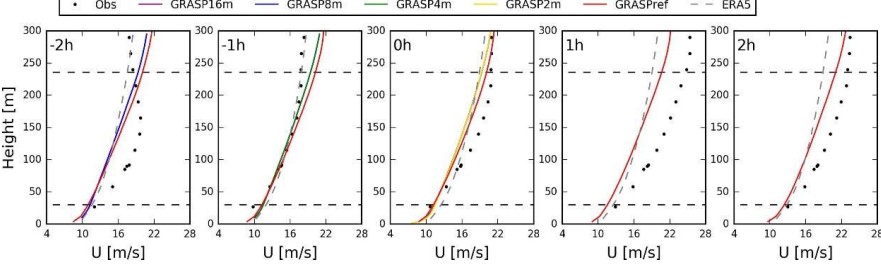

Figure 8. Profiles of wind speed for the wind shear case.

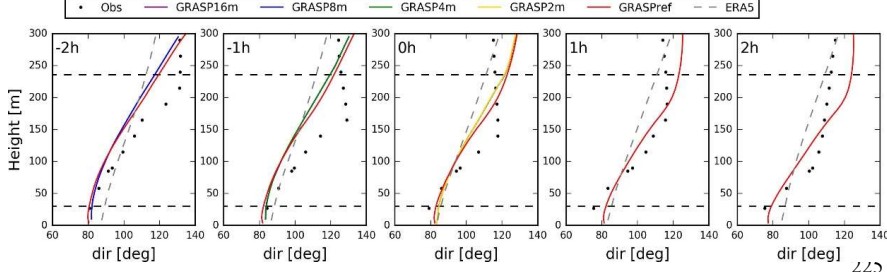

Figure 9. Profiles of wind direction for the extreme veer case.

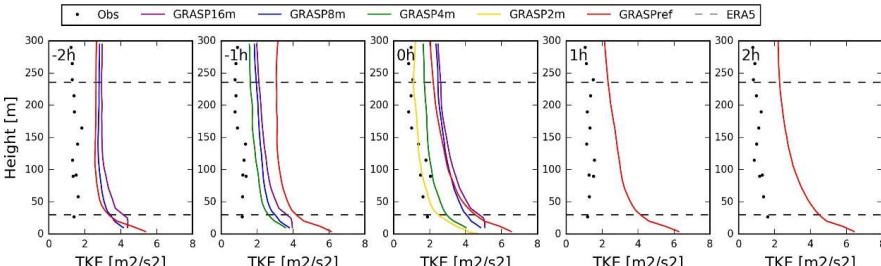

Figure 10. Profiles of TKE for the extreme TKE case.



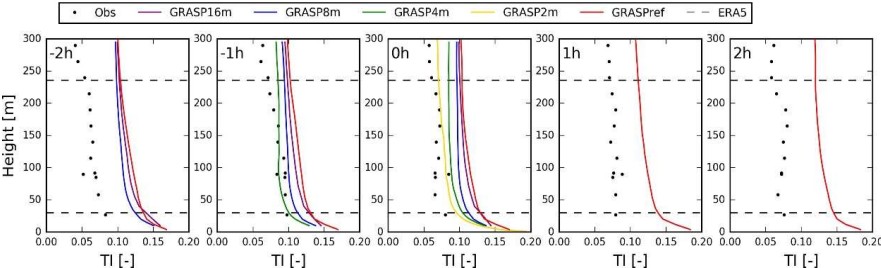

Figure 11. Profiles of TI for the extreme TI case.

**Comparing U, phi, and TI for all five cases**

For t=0, Fig. 12 shows how the variables U, phi (i.e. wind direction), TKE, and TI are connected for the five different cases.
The effect of different GRASP model resolution is small on the mean profiles of wind speed and direction but differences arise in the  turbulence quantities like TI and TKE, where the finest grid-resolution is closest to the measurements. In general, GRASP represents extreme wind shear and veering of wind with height better than ERA5.

It can also be seen that the case with large veer appears to have a strong LLJ, both in the observations and in the model output going together with low values of TI and TKE. Vice versa the case for LLJ could be expected to have significant veer as well.
This is the case in the calculations but much less in the measurements.

In the sequel of the report much attention is paid to the low turbulence intensities  at the selected LLJ event (see table 1), since these low levels affect the loads to a large extent. It is then interesting to note that the results in Fig. 12 show the measured turbulence intensities in the lower part of the rotor plane to be even lower than calculated! On the other hand the measured turbulence intensity in the upper part of the rotor plane is higher than calculated for the LLJ event (not for the other events). It
must be noted however that these values of turbulence intensities come from the  LIDAR which may not measure the turbulence intensity very accurately (Sathe, Gotschall, & Courtney, July 2011). An investigation of the turbulence intensities of the other LLJ events in the year of simulation generally find values of 2% or less (sometimes even less than 1%!) at an altitude of 90 meter   which   is   seen   as   a   confirmation   that   low   turbulence   intensities   are   found   at   LLJ   events   indeed.




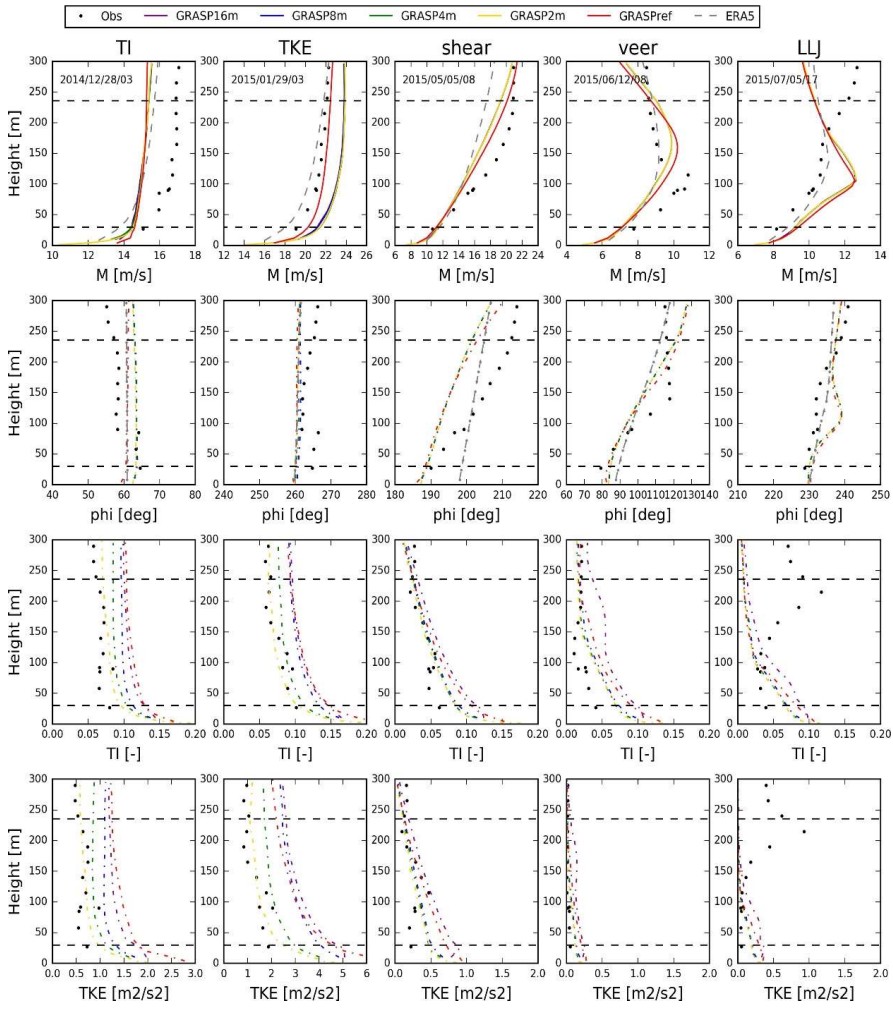


Figure 12. Profiles of U, phi, TI and TKE (rows) for the five selected cases (columns) with high TI, TKE, wind shear, veering, and a strong LLJ.





**Concluding remarks on validation**

In summary, the extreme wind cases that were selected based on GRASP model output, represent 'real weather'. That is to
say, there is a strong qualitative and often quantitative agreement between the modelled and observed extreme events of LLJ,
wind shear, veer, TI and TKE. Although the agreement for the selected LLJ is moderate, it is encouraging to see that many
other LLJ events in the year of simulation find a shear which  is comparable to the measurements. Moreover most LLJ's go
together with low turbulence levels and large veer in both calculations and measurements.

**4.2 Aero-elastic modelling of GRASP extreme events**

The aero-elastic loads in response to the extreme GRASP cases from section 4.1.2. are calculated with the PHATAS code
(Lindenburg, 2005). The development of this code started in 1985 by ECN (now TNO) but later the code has been transferred
to WMC (now LM). The code takes into account blade flexibilities  in all three directions (flatwise, edgewise and torsional)
but also tower and drive train flexibilities. Also the control of the AVATAR turbine is taken into account.

The default aerodynamic solver of PHATAS is based on the Blade Element Momentum (BEM) theory. This is an efficient
but lower fidelity model which, because of its efficiency is used  for industrial design calculations. In its basis such BEM
model is steady and 2D, by which phenomena like yaw and stall are calculated with a very large uncertainty. Therefore, in
the last decades several engineering models have been developed which are added to the BEM theory. These engineering
add-ons cover phenomena like unsteady and 3D effects as well as yaw and stall. They are still of a simplified efficient nature
which makes them suitable for industrial calculations. These engineering models are validated and improved with the most
advanced measurement data (Schepers J. G., November 2012) and with high fidelity models (Schepers J.G. et al, 2018)

Although the default aerodynamic solver of PHATAS is based on the BEM theory, the GRASP events are calculated with a
PHATAS version which is linked to an alternative aerodynamic solver AeroModule as developed by ECN part of TNO.
AeroModule  is a code which has an easy switch between an efficient BEM based model and a high fidelity but time consuming
FVW based model AWSM (Boorsma, Grasso, & Holierhoek, 2012) which allows a straightforward comparison of these two
models with precisely the same input. In this way it can be assessed how well the load response is calculated with a BEM
model in comparison to the load response as calculated from the higher fidelity model AWSM.

In the present study the loads which are considered are the blade root bending moments: flatwise, edgewise and torsion moment
and the shaft moments, torque, tilting and yawing moment on the shaft but the paper limits itself to a discussion of the flatwise
moment only. Generally speaking the conclusions on the flatwise moment are valid for the other loads as well (Schepers, van
Dorp, Verzijlbergh, Baas, & Jonker, 2019)

Both extreme loads and equivalent fatigue loads are considered with a Wohler slope of 10 for the blades and 4 for the shaft.
The loads are calculated in the coordinate system from Germanischer Lloyd.





### 4.3 Interface between GRASP and PHATAS

The input for AeroModule (and so PHATAS) consists amongst others of the 3D wind speeds at several locations in the rotor
plane as function of time. For the present study they were supplied by Whiffle in separate files in NETCDF format in the
resolution which is given in section 4.1.1. They were transformed by ECN part of TNO into TurbSim wind simulator files
(Jonkman, 2009).

It is noted that the turbine yaw angle is fixed and aligned with the time averaged wind direction at hub height from the GRASP
wind input.

**5 Calculation of reference design load spectrum**

The reference design load spectrum for the AVATAR RWT has been calculated and assessed in (Stettner, et al., April 2015).
It is calculated along the IEC standards for wind class IA, which was considered representative for off-shore conditions by the
AVATAR consortium. As mentioned before this is a conservative turbulence class for the present location.

In view of the fact that this deliverable has been produced 3 years before the present study it was decided to repeat the load
calculations with the current version of design tools in order to assure consistency in tools.

The load spectrum from (Stettner, et al., April 2015) is based on an almost complete set of design load cases, i.e. normal
production (DLC 1.2), standstill, stops etc. In the present study it is only the normal production cases from DLC 1.2 which are
repeated. In section 6. it will be shown that these cases are sufficient for the present assessment and there is no need to include
special cases.

The reference load cases are carried out as 10-minute time series for mean wind speeds ranging from 5-25 m/s, with a wind
speed interval of 2 m/s, a shear exponent of 0.2, where the wind input is generated from the stochastic wind simulator SWIFT
using 6 different seeds. The yaw angle is prescribed to be 8 degrees in line with the IEC standards.

It is noted that the aerodynamic model with which the reference spectrum is calculated is based on the default BEM model of
PHATAS where the GRASP events from section 4 are calculated with both BEM and FVW. Apart from fundamental model
differences between BEM and FVW all calculations are carried out in exactly the same way, with the same degrees of freedom,
engineering models used etc in order to assure consistency in results.

### 6. Comparison between aero-elastic loads at extreme events with loads from the reference spectrum

Figure 13 shows the resulting equivalent fatigue flatwise moment as function of the 10-minute averaged wind speed from the
reference design load spectrum and extreme GRASP events. The values indicated with *reference* are the loads as calculated
for DLC1.2. They are compared with the BEM and AWSM calculated loads for the case of extreme Low Level Jet (*LLJ*),
Veer, Shear, Turbulence Intensity (*TI*) and Turbulence kinetic energy (*TKE*).



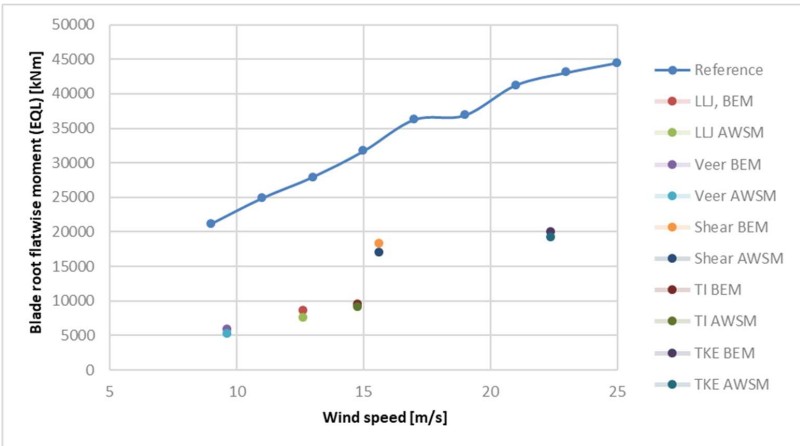

Figure 13: Equivalent blade root flatwise moment: DLC1.2 versus GRASP extreme wind events

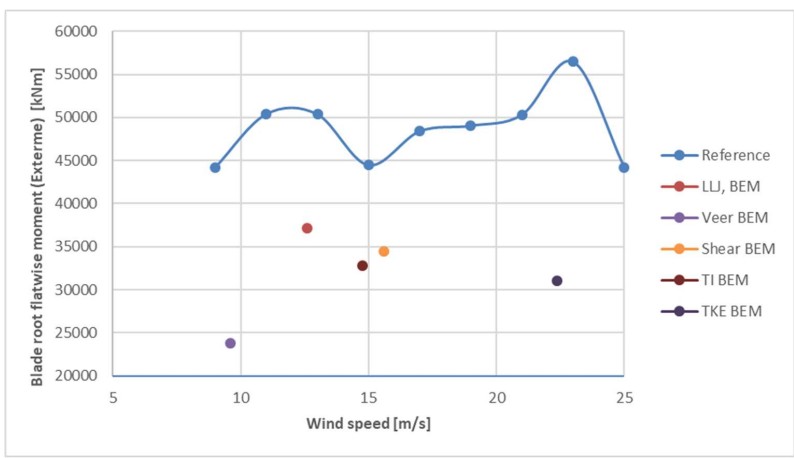


Figure 14: Extreme blade root flatwise moment: DLC1.2 versus GRASP extreme wind events

In Fig. 14 the extreme flatwise moment as extracted from the 10-minute time series is compared and again plotted as function of wind speed. The extreme load has been extracted for a BEM based calculation only. The presentation of extreme loads as function of wind speed is strictly speaking non-sensical since it is the overall maximum value which is determinant. This way

of presenting is chosen because it shows the wind speed where the extremes are found.

The following remarks can be given on the presentation:



- In all cases the extremes were found to be the maximum **positive** values (using the sign conventions from the GL coordinate system).
- The present analysis is based on normal production cases (DLC 1.2) only which means that special and extreme load
cases are excluded. As such the actual maximum extreme load from a full IEC spectrum could even be higher than the values presented in Fig. 14. Some indication for that is found in (Savenije, et al., December 2017) which shows that often non DLC 1.2 cases (e.g. DLC 6.2, idling at storm loads) are more extreme indeed. However, in the sequel it will be shown that even the extreme loads from DLC 1.2 are higher than the loads from the extreme GRASP events by which there was no use to calculate the non-DLC1.2 load cases.
- The design load spectrum has been calculated for 3 different seeds per wind speed. The results from Fig. 13 is based on the averaged equivalent load. The values from Fig. 14 values are the overall extremes per wind speed.

The most important conclusion is that the loads in response to the extreme wind events from GRASP remain within the load envelope of the reference spectrum. This is true for the equivalent fatigue loads, see Fig. 13, which shows that all EQL's from the GRASP extreme events are lower than the EQL's from the reference DLC 1.2 at comparable wind speeds. It is also true

for the extreme loads, see Fig. 14. As explained above the ''real" extreme reference loads are likely to be even higher than the values given in these figures, since the results in these figures consider DLC 1.2 only. This makes that the extreme loads from the GRASP wind events remain even more within the reference spectrum.

Another important conclusion is that the EQL of the blade root flatwise moment is overpredicted with the BEM model (assuming that the fatigue loads as calculated with the FVW model AWSM are close to reality). Similar observations were

made in (Boorsma, Chasapogianis, Manolas, Stettner, & Reijerkerk, September 2016) and (Boorsma, Wenz, Aman, Lindenburg, & Kloosterman, September 2019) where differences are reported in the order of 10-20% for load cases which are representative to IEC normal production.  The present study shows overpredictions which are in the same order of magnitude i.e. 14% for the extreme LLJ, 11% for the extreme veer case, 7% for the extreme shear case but only 4-5% for the extreme turbulence intensity and turbulent kinetic energy. The difference between AWSM and BEM based fatigue shaft loads (not

shown in this paper) were generally found to be smaller and less straightforward than for the blade root flatwise moment: in some cases AWSM even predicts higher fatigue loads than BEM.

In order to gain some further understanding on the results, the loads from the Low Level Jet are analyzed in more detail. Thereto table 2 presents the EQL of the flatwise moment for the Low Level Jet from BEM and AWSM in the third and fourth row respectively. In the second row the corresponding EQL from DLC1.2 is given for a wind speed of 13 m/s which is very

close to the 10-minute averaged hub height wind speed at the Low Level Jet. In the second column the EQL is calculated for a flexible construction (which correspond to the results from Fig. 13). The results in the third column give the EQL for a rigid construction (these are calculated for the Low Level Jet only, not for DLC1.2).

The fourth column gives the EQL from the azimuthally binned averaged  variation. This azimuthally binned averaged variation is (for a linear system) similar to the deterministic variation which is mainly a result of the shear (although the veer

in the LLJ event and the 8 degrees yaw error for DLC 1.2 leads to a deterministic variation as well). The equivalent loads from the deterministic variation are calculated for the BEM results only.






|  | M$_{flat, EQL}$ Flexible [Nm] | M$_{flat, EQL}$ Rigid [Nm] | M$_{flat, EQL}$ deterministic [Nm] |
|---|---|---|---|
| DLC1.2(BEM), 13 m/s | 27955000 | | 15904000 |
| LLJ(BEM) | 8661000 | 11163000 | 8584700 |
| LLJ(AWSM) | 7602100 | 9461100 | |

Table 2: Comparison of equivalent blade root flatwise moment for extreme low level jet

- It can be observed (as already noted from Fig. 13) that the equivalent flatwise moment at the LLJ is only 31% (approximately) of the equivalent load from DLC 1.2. Some explanatory remarks can be made:
    o As mentioned in section 4.1.2 the turbulence level at the Low Level Jet is extremely low (approximately 1.6 %
at hub height) where the turbulence level for DLC1.2 at 13 m/s is in the order of 19%. The very low turbulence level at the LLJ explains, at least partly, the much lower fatigue load. This is confirmed by the EQL of the deterministic variation in the third column which is almost similar (99%) to the EQL of the total variation in the first column. The 1% difference is the addition from turbulence and should be compared with the difference between deterministic and total variation from DLC 1.2 which is approximately 43%. This indicates how little
the low turbulence level at the LLJ adds to the fatigue loads.
    o Still the EQL of the deterministic variation at the LLJ is much lower (approximately 54%) than the EQL of the deterministic variation at DLC 1.2. This indicates that the low fatigue loads at a LLJ are not only caused by the lower turbulence levels but it is also the different shear from the LLJ which lowers the EQL. Some further explanation is offered by Fig. 15. This shows a comparison between the azimuthally binned averaged flatwise
moments for the LLJ and DLC1.2. Azimuth angle zero indicates the 12 o clock position, and the rotor rotates clockwise so azimuth angle 90 indicates the 3 o clock position when looking to the rotor. The variation from DLC 1.2 has a 1P variation with a relatively large amplitude. This is the behavior of the flatwise moment in an atmosphere with 'common' vertical wind shear. The wind speed (and so the loads) decreases when the blade rotates from the vertical upward 12 o'clock (zero azimuth) position to the vertical downward 6 o'clock (180
azimuth). It then increases again when the blade rotates from 180 degrees towards 360 degrees.
    o The variation from the low level jet is very different. It leads to a 2P variation with a relatively small amplitude. This 2P variation can be explained with the wind speed variations from Fig. 2. At zero degrees azimuth (the 12 o' clock position, when the blade is pointing vertical upward) the low wind speed at maximum altitude leads to a low load. When the blade rotates to the 3 o' clock position (90 degrees
azimuth) the high velocity at hub height increases the load. A further rotation towards 12 o' clock (180 degrees) decreases the load due to the low wind speed at the lowest position. Finally a rotation towards 270 degrees increases the loads as a result of the higher wind speed at hub height.

    In section 4.1.2 it was mentioned that the shear from the LLJ in the lower part of the rotor plane (between
90 and 270 degrees) is very comparable to the shear with exponent 0.2 which is assumed in DLC1.2. This is confirmed by a more or less similar slope dM$_{flat}$/d$\phi_r$ for DLC1.2 and LLJ in the lower part of the rotor plane, i.e. from $\phi_r$ is 90 to 270 degrees with $\phi_r$ the azimuth angle.

    Eventually the lower load amplitude from the LLJ leads to a lower fatigue load than the fatigue load from
DLC 1.2, even though these variations happen twice as often.

    It is noted from Fig. 2 that the present LLJ has a maximum velocity close to hub height and it could be





argued that a different hub height leads to a different load behavior. The lowest part of the rotor plane of the AVATAR RWT is at an altitude of 29.8 meter and the upper part is at an altitude of 235.6 meter. It was not considered feasible to decrease the tower height and lower the rotor plane even more. Also a lowering of hub height would bring the maximum in LLJ wind speed even closer to hub height (See Fig. 2). Therefore an increase of tower height has been investigated but this was limited by the domain size of the GRASP field which extends up to a maximum altitude of 255 meter. Hence the tower height can not increase with more than 19.4 meter.  A hub height of 250.7 meter has been investigated but this did not

lead to significantly different conclusions (i.e. the loads from the LLJ remain within those of the reference spectrum)

- As explained before BEM overpredicts the EQL with approximately 14% compared to the EQL from FVW.   For the rigid construction however the difference is even larger, i.e. 18%. The commonly believed explanation for the overpredicted BEM EQL lies in a more local tracking of the induced velocity variations in FVW models, by which they

vary synchronously with the variation in inflow. This synchronization then damps out the variations in angle of attack. Moreover, FVW models allow for a more intrinsic and realistic modelling of shed vorticity variations in time. The lower differences for a flexible construction could be a result of the flexible blades being bend by wind speed variations and so leading to smaller angle of attack variations.

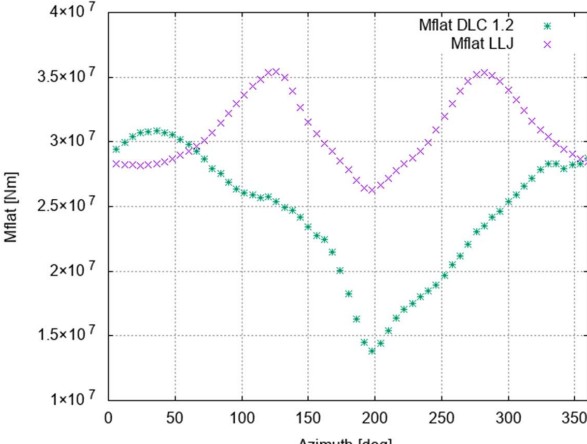


Figure 15 Azimuthally binned averaged flatwise moment: LLJ versus DLC1.2

## 6 Conclusions and recommendations

- A successful coupling has been established between the LES wind field model GRASP from Whiffle and the aero-

elastic code PHATAS (with AeroModule) from ECN part of TNO. Thereto extreme events, including a Low Level Jet are selected from a 1 year simulation of GRASP wind fields. These events are fed as wind input files to the PHATAS code and used to simulate the AVATAR 10 MW Reference Wind Turbine (RWT) at an off-shore location.



- A validation of the LES Wind fields has taken place by comparing the calculations with measurements from Meteorological Mast IJmuiden. This validation shows that there is generally a good agreement in the load determining characteristics of the LES wind fields by which the calculated events can be used with confidence to assess the importance of them in an aero-elastic load context. However more validation is needed, in particular on turbulence characteristics at high altitudes (say higher than 100 meter)

- The resulting (EQL and extreme) loads for the selected events are (roughly speaking) 30-70% lower than those from the reference design load spectrum of the AVATAR RWT. As such, the often heard expectation that low level jets have significant impact on loads is not confirmed for the present off-shore situation. This is partly explained by the very low turbulence intensities which go together with the LLJ. However the deterministic EQL from the LLJ shear is also lower than the deterministic EQL from DLC 1.2. This is due to the fact that the shear from the LLJ is not very extreme in comparison to the shear from the IEC standards.  The LLJ shear profile then leads to a 2P variation instead of a 1P variation from 'normal shear' but the amplitude is smaller resulting in a lower fatigue damage. From the results one could hypothesize that the combination of the shear and turbulence levels from the IEC standards may often lead to conservative loads. However much more research is needed to warrant a conclusion, especially in the validation of the on-site turbulent wind fields.

- It is noted that the present LLJ has, more or less by coincidence, a maximum velocity close to hub height. A study on different hub heights didn't show a very different outcome but the limited domain size of the LES wind field made that the hub height could not increase with more than 20 meter. A study with a much taller tower (and so an extended domain size)  is recommended.

- For the selected extreme events the EQL from the more physical AWSM model are considerably lower than the EQL of BEM model which indicates that BEM overpredicts fatigue loads. The difference is largest for the shear driven cases and for a rigid construction. Efforts should be undertaken to improve the BEM fatigue calculations for such shear events.

- The present research can be considered as a 'pilot' study to investigate the potential of a coupling between turbine response models and high fidelity wind models. The success of it leads to the recommendation to explore such coupling even further for the calculation of a full design load spectrum. This makes it possible to  assess the validity of a conventional method for the calculation of a design load spectrum based on stochastic wind simulators. The higher fidelity of the present method makes that eventually design calculations could be based on physical wind models.

- Although the coupling between PHATAS and GRASP was very successful the interfacing through GRASP output and PHATAS wind input files can be improved. Ideally an integrated approach should be developed without the need of interface files.

*Author contributions*. J.G. Schepers assembled and ran the load simulation results and analysed the overall results. P. van Dorp and H.J.J. Jonker modified the LES code and performed the GRASP simulations. R.A. Verzijlbergh assisted in the analysis of the results. P. Baas performed the validation of the GRASP simulations

*Acknowledgement* The research was sponsored by the Topsector Energy Subsidy from the Ministry of Economic Affairs and Climate; F. Savenije (ECN part of TNO) is acknowledged for the calculation of the reference load spectrum. K. Boorsma (ECN part of TNO) is acknowledged for his support on the AeroModule code.

*Competing interests*. The authors declare that they have no conflict of interest.



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
