# Peer review of "Aero-elastic loads on a 10 MW turbine exposed to extreme events selected from a year-long Large-Eddy Simulation over the North"

_Wind Energy Science, 2020_

## Referee Comment (RC1) · Anonymous Referee #1 · 31 Mar 2020

Dear authors, Thank you for the submission. To start, please note that the submission does not respect the formatting guidelines of the Wind Energy Science Journal. This clearly jeopardizes readability. Among the many changes you will be required to implement before publication, please avoid multi-level indentations and check the spacing of the text, which in the current submission varies from paragraph to paragraph. In addition, be consistent with the style of the figures, and most importantly use vectorized images. Figures 6-12 are poorly readable, and zooming in does not help. Overall, the formatting of the submission substantially reduces the quality of the work.

On the technical side, my impression is that the paper blends two interesting topics

into a not always clear discussion. You are mixing differences in the wind prescribed by the standards (wind from a met mast and wind from a high-fidelity wind solver) with a comparison between BEM and vortex models. Simultaneously, you ran simulations with flexible and rigid rotor blades. Overall, I find the results and the discussion confusing: I do not understand any more what comes from what. I strongly recommend distinguishing better the effects of each analysis. The two analyses (different wind inputs, BEM vs vortex) can stay in the same paper, but not as they are right now. Please isolate the effects and structure the discussion more clearly.

In addition, please address these comments: 1) I do not agree with the authors that the comparison between the observations at MMIJ and the predictions of GRASP is found satisfactory. I find the matching between the black dots and the colored lines quite poor. Please elaborate better about these differences. I don't really understand where the differences come from. Several potential sources are mentioned, but I get lost and I cannot draw any conclusion. Also, how can you later distinguish between the colored lines? How do you use them in the load analysis? 2) Let's for example look at Figure 6. The match at 0h is very poor. Why is it better at 2h? How do you use this discrepancy in the load analysis? What is the point of using GRASP if the matching is so poor? 3) Figure 10 shows an especially poor matching between numerical and field results. Please discuss the origin of this discrepancy. 4) Please make sure to spell out the acronyms. What is the EQL? Is it the damage equivalent load? If that is the case, DEL is much more commonly used than EQL. 5) At page 15 line 318 you write "In section 6. it will be shown that these cases are sufficient for the present assessment and there is no need to include special cases.". First, please note that you have two sections 6. Next, the paper has no analysis about the "special cases", so the sentence above is not really supported by the results. 6) At Page 17 line 344 you write "The present analysis is based on normal production cases (DLC 1.2) only which means that special and extreme load cases are excluded. As such the actual maximum extreme load from a full IEC spectrum could even be higher than the values presented in Fig. 14. Some indication for that is found in (Savenije, et al., December 2017) which shows that often non

DLC 1.2 cases (e.g. DLC 6.2, idling at storm loads) are more extreme indeed. However, in the sequel it will be shown that even the extreme loads from DLC 1.2 are higher than the loads from the extreme GRASP events by which there was no use to calculate the non-DLC1.2 load cases." I find this sentence confusing, please consider revising it. Are you saying that DLC 1.2 does not generate ultimate loads? Anyone who runs load analysis of wind turbines knows that most of ultimate loads do not originate from normal operating conditions, but from either extreme turbulence (1.3), extreme change of direction (1.4), faults (2.x), or storm cases (6.x). Your considerations are not at all a surprise, and I do not think that you need a full paragraph to state that. Also, please distinguish between 1.1 (ultimates) and 1.2 (fatigue). 7) My understanding is that you are feeding to the aeroelastic solver wind histories characterized by much lower turbulence intensities than standard class IA conditions. It is therefore not surprising that the maximum and fatigue loads are lower. Please elaborate about this. What conclusions can you draw after running the full comparison? 8) Please elaborate about the dynamic controller. The only detail I find is "Also the control of the AVATAR turbine is taken into account.", with no references. This would force readers to dive into the deliverables of INNWIND, which are neither few, nor always peer-reviewed. 9) Are you sure that no other discrepancies between the models affect your results? 10-20% difference in fatigue loads between BEM and vortex for a rigid rotor configuration seems excessive to me. The work https://www.wind-energ-sci-discuss.net/wes-2020-6/wes-2020-6.pdf, shows some differences, but it looks to me that they are far smaller than the ones reported in this work. Is the sheared inflow the source of the discrepancy? If you confirm the result, please discuss the sources of the difference. Also, please separate more clearly the comparison between 1.2 and LLJ, and the comparison BEM vs vortex. I got very lost while reading section 6. Why not splitting it in subsections 6.1 "1.2 vs LLJ", and 6.2 "BEM vs vortex"? 10) What is the last column of Table 2 "Mflat, EQL deterministic [Nm]". I don't understand what deterministic refers to. Please clarify. 11) The paragraph discussing the difference between the physical reason behind the 1P and 2P harmonics is interesting, but hard to read. To start, please do not structure

it in a two-level bulleted list. Next, please clearly separate the comparisons between modeling fidelity and wind inflow. 12) Please distinguish clearly between final conclusions and recommendations for future work. It appears to me that the conclusion of this work are not strong nor novel. Much seems instead work in progress. This is a strong shortcoming.
* * *

---

## Referee Comment (RC2) · David Verelst (Referee) · 8 May 2020

Thank you for your contribution. This work touches upon a wealth of different challenging topics, all of which are important within the context of wind turbine loads.

Some general remarks:

- Although the work is very relevant, the way it is presented feels a little rough around the edges. This work could benefit from an editorial focussed review.

- The difference between the measurements of inflow fields and the reconstructed

fields still shows some reasonably large differences. How would you expect the difference between the measured and simulated fields to affect the loads?

- The authors touch three very big area's: high fidelity measurement campaign with lidar and met mast, BEM vs vortex models, extreme and fatigue loads for a range of inflow conditions. In between the authors also refer several times to a "sequal report". A more careful structuring of the text and avoiding referring to work that seem to be completed yet but is not included in the manuscript (why, due to space concerns?) might help avoiding the reader getting confused since so many different topics are already considered.

- I very much appreciate the efforts of the authors to identify various real and complex inflow conditions compared to the standards, and especially also the consideration that BEM with disk averaged induction needs to be considered carefully in that context.

Some specific comments:

- line 85: do you mean 365 simulations of 24 hours each, and as such arrive to 1 year?

- line 91: could you indicate how the azimuthal resolution compares to the standard IEC turbulence box resolution?

- line 91: could you also give an indication of how expensive these simulations are in terms of computational time and resources?

- Figure 3 and Figure 4 are not referred to in the text.

  – include reference to figure 2 also on line 101 and further around line 110?

  – should fig 3 and fig 4 be referred to around line 110-130?

- figure 6: to which grid size is the label GRASPref referring to?

- paragraph at line 180: I am not sure I understand this, are you referring to results from a study you did but have not published yet? Why not include those results, who show better agreement, here instead? That also begs the question, what causes some cases to match better than others?

- line 185: what is an Excel polynomial curve fitting? I assume you just have used an n-th order polynomial to fit something by minimizing a least-square cost function or something?

- lines 245-250: it is a very interesting teaser to this sequel report, but why not include the results here? As a reader I get the impression the authors already done the necessary analysis.

- figure 12 is confusing to me, it took me a while to figure out that TI, TKE, shear, ... on the titles of the upper row referred to the "TI: case at which extreme TI has been observed". At first this is not obvious because TI, TKE are the same labels used on the x-axis of the respective TI, TKE row plots.

- figure 13: it is clear that the observed 1Hz equivalent loads (EQL) are lower, but strictly speaking that is only half the story. The other half is the frequency/distribution at which all the different events occur and together they result in the life time equivalent load. You could consider pointing that out as well.

Regarding: 6. Comparison between aero-elastic loads at extreme events with loads from the reference spectrum:

I assume the extreme loads from DLC1.2 are based on at least 6 seeds per wind speed at 3 different yaw inflow angles (-8, 0 and 8 degrees)? In contrast, only 1x 10 minute realisation is used for the extreme events since they are directly based on

a reconstructed/measured inflow field. I understand that in this context the authors want to demonstrate that extreme inflow conditions based on observations/high-fidelity simulations does not necessarily lead to higher loads when compared to the standards. I agree that is an important observation. However, I think it remains plausible that you could in theory create very similar inflow conditions as you have observed leading to extreme loads exceeding the DLC1.2 reference values. Since there is so much uncertainty and variation in turbulence levels across the rotor plane with smaller local "bursts" driving an outlier event (for example: https://dx.doi.org/10.1002/we.497). The more simulations you consider, the higher the likelihood you bump into such an event. From that perspective comparing extremes is only sensible when considering many 10 minute realisations. You could consider stating even stronger in the conclusions that many more aspects require more analysis, and that this study contributes to that bigger picture.

---

## Author Comment (AC1) · 30 Jun 2020

We thank both reviewers for a thorough and critical review and for rightfully pointing out several points where our manuscript can be improved.

Although both reviewers use slightly different words we think that they both address some common criticism. In this document we will start with explaining our answers to this common criticism.

Thereafter we will discuss the remaining specific comments from each of the reviewers.

**Common comments**

- The agreement between calculated and measured wind data is not considered convincing. We understand this criticism, see Appendix A.
- The paper discusses too many unrelated subjects by which the reviewers get confused. Yes we understand this criticism. We want to overcome it with the strategy explained in Appendix B
- Not all conclusions are definite, some of the research is still work in progress. Our answer to that is found in in Appendix C

**Specific comments from reviewer 2**

We appreciate the fact that reviewer 2 acknowledges the relevance of our work in general and our efforts to identify various real and complex inflow conditions compared to the standards for load calculations.

General comments: Most of them are covered above. Moreover we will not refer anymore to the sequel report with additional results. These are just more of the same and all references to them led to confusion. You are right that these results were excluded due to space constraints but we can do without them.

Line 85: Yes, we have performed 365 simulation of 24 hours (plus a 2h spin-up period for each simulation).

Line 91: An azimuthal interval of 6 degrees is what we often use in our aero-elastic simulations indeed.

Line 91: The computation time of the LES runs amounts to roughly 2 days on a cluster with 4 NVIDIA Volta GPUs. Computation time of the load calculations is much faster than realtime for BEM on a simple laptop. The Free vortex wake calculations are a factor 100-1000 slower (dependent on number of wake points and the wake cut-off length etc).

Figures 3 and 4 are implicitly mentioned on line 106 but it is much better to follow your suggestions and refer to the specific figures indeed. We will do that in the revision

Figure 6: Graspref: This figure will be removed, and we will discuss the validation along the lines given in Appendix A.

Line 180: What we were trying to say is that we found a relatively poor agreement between calculations and measurements for the strongest LLJ from figure 6 but the other LLJ cases from the same year are predicted better, see figure 7. We could reformulate it follows: *Figure 6 shows a rather poor agreement for this particular LLJ event. However there are other LLJ events which are predicted much better, see figure 7.*

However, in Appendix A it is explained that we will describe the validation in a different way and the figure will not be included in the revisions.

Line 185 Excel polynomial fitting: Actually this is a built in smoothing function (*indicated through scatter with smooth lines*) which is not documented by MS but appears to be some kind of Catmull Rom spline. In the revision the figure will be deleted anyhow

Line 245-250. These sentences might confuse the reader indeed. What we were trying to say is that the low turbulence intensities from GRASP which go together with a LLJ are largely responsible for the low loads where these low turbulence intensities are believed to be true. *We will reformulate it as follows:*
In the sequel of the report it will be shown that the loads from the LLJ are relatively low. The low loads at a LLJ are partly caused by the very low turbulence intensities which go together with an LLJ. It is then important to know whether these low turbulence intensities at LLJ's are also found in the measurements (as a matter of fact the measured turbulence intensities are even lower).

Figure 12: Yes, you are right but we will eliminate this figure.

Figure 13: Yes I fully agree. Using an EQL's conceal the underlying RFC and frequency information. We will explain that in the revision

Regarding 6: Your assumptions are right: We used 6 seeds for the three different yaw angles. We will point that out in the revision. And yes more realizations are needed to get a better picture of the extremes. We will also add this to the conclusions.

**Comments from reviewer 1**

General

- Messy formatting. Our sincere apologizes, you are very right. Somehow Word is not doing what it is supposed to do but we will solve it. Also apologizes for chapter 6 which occurs twice indeed. The chapter number for the conclusions should be 7
- Two different topics in one paper. See Appendix B

1) 2) and 3) Agreement between calculations and measurement is not so good. See Appendix A

4) Acronyms. Apologies and yes we will spell them out in the revision. EQL is the equivalent load indeed. I remember this abbreviation was often used in the 1980's when this concept was introduced. Nowadays DEL may be more commonly used indeed.

5) and 6) Yes you are fully right, the ultimate loads generally do not come from the DLC1.2. What we are trying to say is that the ultimate loads from the extreme GRASP cases are lower than those from the load set based on DLC1.2 only. But then they should definitely be lower than the ultimate loads from the full IEC load set because the full spectrum considers more cases and the ultimate loads can only become higher by adding load cases.

7) this is indeed what we are trying to say. But it is the LLJ (and extreme shear) which give this low turbulence intensity. The fact that a LLJ and extreme shear give a low turbulence intensity is in itself not so surprising because we know that a high shear goes together with a low turbulence level. But it is important to realize this low turbulence intensity at LLJ's goes together with a low turbulence

level by which the loads are reduced. As mentioned above,  LLJ's are often considered negative in terms of loads.  We are planning to add another picture that shows that both GRASP and the observations have similar TI - LLJ statistics.

8) You are right that details on the control are not given. The controller is believed to be representative since it has been designed with standard control design tools available at the partners in the INNWIND.EU and AVATAR projects with which they often design controllers for industry. Below rated wind speed, the turbine controller aims for maximum power production with variable rotor speed operation using a speed dependent generator torque setpoint (for optimum tip speed ratio) and constant optimal blade pitch angle. Above rated wind speed, the rotor speed and generator power are regulated to their nominal rating using constant generator torque and collective blade pitch control. We will include this information in the revision

9) We use the AeroModule which is a single code with an easy switch between the two different aerodynamic models: BEM and FVW. This assures the rest of the input (geometrical, structural, aerodynamic blade data, turbine data, control algorithm) to be the same. So yes we are sure that the differences are caused by the aerodynamic models. They are consistent to conclusions from the AVATAR project and although differences are large indeed the 14% difference is not larger than the differences found in the  AVATAR project. In the final report from AVATAR it is written that:
*Comparisons between aero-elastic calculations based on BEM showed a 15% over-prediction in fatigue loads compared to those from FVW, probably due to an inaccurate prediction of time-varying induction effects.*
See also Appendix C

10) The deterministic component is the (mainly) the shear driven component. In the revised version of the paper we will call it the azimuthally binned averaged component, which in the case of a linear system is similar to the deterministic component.

11) yes I found it difficult to explain the 1P and 2P behavior. I will reformulate it as follows which I hope is better understandable

*…. Some further explanation is offered by Fig. 15. This shows a comparison between the azimuthally binned averaged flatwise moments for the LLJ and DLC1.2. Azimuth angle zero indicates the 12 o clock position. The rotor rotates clockwise so azimuth angle 90 indicates the 3 o clock position when looking to the rotor. The variation from DLC 1.2 has a 1P variation with a relatively large amplitude. This is the behavior of the flatwise moment in an atmosphere with 'common' vertical wind shear. The wind speed (and so the loads) decreases when the blade rotates from the vertical upward 12 o'clock (zero azimuth) position to the vertical downward 6 o'clock (180 azimuth). The flatwise moment increases again when the blade rotates from 180 degrees towards 360 degrees.*

*The variation in flatwise moment from the low level jet is very different. It shows a 2P variation with a relatively small amplitude. This 2P variation can be explained with the LLJ wind speed profile from Fig. 2 which shows the wind speed to be low at 0 degrees azimuth  (the 12 o' clock position, when the blade is pointing vertical upward) and at 180 degrees (the 6 o' clock position, when the blade is pointing vertical downward). The wind speed is maximum at (approximately) hub height which correspond to azimuth angles of 90 and 270 degrees  (i.e. the 3 o' clock and 9 o' clock position when the blade is standing horizontally)*
*This velocity variation is reflected in the flatwise moment. It is low at 0 degrees, high at (roughly) 90 degrees and 270 degrees and low again at 180 degrees.  This leads to a 2P variation but the  load amplitude is relatively small. Hence although the 2P load variation happens twice as often as the 1P load variation from the DLC 1.2. the lower amplitude of the variations lead to a lower fatigue*

11): See Appendix C.

**Appendix A: Agreement between calculations and measurements**

We agree that the correspondence of a some of the presented 10-minute average wind speed profiles may be considered poor considering typical yearly averaged error metrics (modeled vs. observed wind speed) of a mesoscale or reanalysis dataset. However, we would like to point out that this can be expected when focusing on extreme cases, which are, by definition, the outliers in a dataset. The reviewer comments have made us realise that, in fact, a one-to-one comparison of extremes is not the most relevant. What is more relevant is the question: does GRASP capture, in a climatological sense, the (extreme) wind characteristics? In our revision of the paper, it is exactly this approach that we intend to take: focus on the validation on the distribution (occurrences and magnitudes) of extreme wind events rather than a point-by-point comparison of single extreme events.

However, the reviewer comments were already indicating that the paper is trying to convey two messages in one paper. So, although an elaborate validation of extreme winds is an important and necessary next step that needs to be performed and presented in detail, we feel that putting too much emphasis in this would create a severe imbalance in the contents of the paper. To find the right balance between the validation of GRASP winds and the actual aero-elastic load calculation, we intend to include a number of extra figures showing the distributions of wind and turbulence and 90th percentiles of strong veer and shear events and to remove the one to one comparisons of the profiles. Furthermore, we intend to give more emphasis to further GRASP validation in our recommendations for future work. As an example, we include a few of the plots we intend to include in the revision below. Note that these are preliminary and not the final format or quality.

[Figure]

Fig: Scatter density plot of modeled vs. observed 92m wind speed.

[Figure]

Scatter density plot of modeled vs. observed TI at 92m height.

[Figure]

Comparison of the 90th percentile strongest shear and veer conditions from observations, ERA5 and GRASP.

[Figure]

Standard deviation of the wind speed versus wind speed at 92 m for one year of MMIJ data from observations (left panels) and GRASP (right panels). Top panels represent all data, bottom panels only include data records for which either the observed or the modeled wind speed profile is classified as low-level jet (5.6% of the data). For reference, the solid lines indicate TI values of 10% (upper lines) and 5% (lower lines)

**Appendix B: Too many messages in one paper.**

The paper addresses too many messages which confuses the reader.

Yes we understand and that is why we had added section 2.

In section 2 it is mentioned that, apart from demonstrating the combined GRASP-PHATAS tool, the research aims to investigate the impact of extreme events on the load spectrum but also the accuracy of a standard BEM model for the calculation of these events. These are two different subjects indeed which are to some extent unrelated. However, in our case the second subject is a spin-off from the first subject in view of the fact that we performed our calculations with

AeroModule which is a code based on two different aerodynamic models: A BEM based model and a higher fidelity FVW model. So insights on the accuracy of BEM are automatically obtained. We then feel it is a pity not to share these insights and this is supported by the comment from the second reviewer who writes that the consideration of BEM with disc averaged induction needs attention.

To make a long story short: We prefer to keep most of the results in the paper but we will prepare the reader in section 2 better on the fact that results are discussed on two unrelated subjects by explicitly mentioning this.

Another step with which we will improve readability is to divide section 6 in two subsections:

- o Assessment of loads from extreme events
- o Accuracy of calculating loads from extreme events.

This is in line with the recommendation from the first reviewer

In order to improve readability we will also skip the result for a rigid rotor and the discussion of flexibility effects. In retrospective we believe that these results and the associated discussion are less relevant and they only add to the confusion.

**Appendix C Conclusions are preliminary and not new:**

Our main take away messages are:

- The extreme events and in particular the LLJ's which we have considered donot give loads which exceed the loads from a standard design load spectrum. We think this is new because it is often stated that LLJ's have a negative effects on loads, see e.g. Duncan, J. November 2018 . But it is a preliminary conclusion indeed. We agree that the methodology needs further validation and we need to consider more aspects (more seeds, more years etc). We tried to make that clear by saying *that more research is needed to warrant a conclusion, especially in the validation of the on-site turbulent wind fields*. So we agree that this is a preliminary conclusion indeed but we are not sure how to make this more explicit.

- BEM overpredicts fatigue loads for shear driven cases. We are confident on this conclusion. Indications for this overprediction have also been found in other studies, we refer to the EU project AVATAR and also the recently finished Dutch national project Vortex Loads (Boorsma, K., Wenz, F., Aman, M., Lindenburg, C., & Kloosterman). However both AVATAR and VortexLoads consider artificial load cases where the present study is based on extreme cases which result from a physical LES model. So we think this is a new conclusion.

---

## Author Response (AR1)

We thank both reviewers for a thorough and critical review and for rightfully pointing out several points where our manuscript can be improved.

Although both reviewers use slightly different words we think that they both address some common criticism. In this document we will start with explaining our answers to this common criticism.

Thereafter we will discuss the remaining specific comments from each of the reviewers.

**Common comments**

- The agreement between calculated and measured wind data is not considered convincing. We understand this criticism, see Appendix A.
- The paper discusses too many unrelated subjects by which the reviewers get confused. Yes we understand this criticism. We want to overcome it with the strategy explained in Appendix B
- Not all conclusions are definite, some of the research is still work in progress. Our answer to that is found in in Appendix C

**Specific comments from reviewer 2**

We appreciate the fact that reviewer 2 acknowledges the relevance of our work in general and our efforts to identify various real and complex inflow conditions compared to the standards for load calculations.

General comments: Most of them are covered above. Moreover, we will not refer anymore to the sequel report with additional results. These are just more of the same and all references to them led to confusion. You are right that these results were excluded due to space constraints, but we can do without them.

*Line 85: do you mean 365 simulations of 24 hours each, and as such arrive to 1 year?*

Yes, we have performed 365 simulation of 24 hours (plus a 2h spin-up period for each simulation). This has been added to the text.

*Line 91: could you indicate how the azimuthal resolution compares to the standard IEC turbulence box resolution?*

An azimuthal interval of 6 degrees is what we often use in our aero-elastic simulations indeed. This has been added.

*Line 91: could you also give an indication of how expensive these simulations are in terms of computational time and resources?*

The computation time of the LES runs amounts to roughly 2 days on a cluster with 4 NVIDIA Volta GPUs. This has also been added.
Computation time of the load calculations is much faster than real-time for BEM on a simple laptop. The Free vortex wake calculations are a factor 100-1000 slower (dependent on number of wake points and the wake cut-off length etc).

*Figure 3 and Figure 4 are not referred to in the text.*

*– include reference to figure 2 also on line 101 and further around line 110?*

*– should fig 3 and fig 4 be referred to around line 110-130?*

These figures have been removed and replaced by a new figure that shows all the relevant case profiles in one figure with several panels.

*Figure 6: to which grid size is the label GRASPref referring to?*

This figure has been removed, and we will discuss the validation along the lines given in Appendix A.

*Paragraph at line 180: I am not sure I understand this, are you referring to results from a study you did but have not published yet? Why not include those results, who show better agreement, here instead? That also begs the question, what causes some cases to match better than others?*

What we were trying to say is that we found a relatively poor agreement between calculations and measurements for the strongest LLJ from figure 6 but the other LLJ cases from the same year are predicted better. However, in Appendix A it is explained that we will describe the validation in a different way and the figure has not been included in the revisions. Instead, we have presented the average low-level jet profiles over the year for the observations, GRASP and ERA5.

*Line 185: what is an Excel polynomial curve fitting? I assume you just have used an n-th order polynomial to fit something by minimizing a least-square cost function or something?*

Actually, this is a built-in smoothing function (*indicated through scatter with smooth lines*) which is not documented by MS but appears to be some kind of Catmull Rom spline. In the revision the figure is deleted anyhow

*Lines 245-250: it is a very interesting teaser to this sequel report, but why not include the results here? As a reader I get the impression the authors already done the necessary analysis.*

These sentences might confuse the reader indeed. What we were trying to say is that the low turbulence intensities from GRASP which go together with a LLJ are largely responsible for the low loads where these low turbulence intensities are believed to be true.
We have revisited this entire section. We have included a figure (new version figure 6 bottom panels) that shows the wind speed standard deviation against wind speed average for the observations and GRASP conditioned on the occurrence of a LLJ. This figure shows that LLJ indeed coincide with low TI levels and that observations and GRASP are in agreement about this. We trust this whole point is now presented much clearer.

*Figure 12 is confusing to me, it took me a while to figure out that TI, TKE, shear, ... on the titles of the upper row referred to the "TI: case at which extreme TI has been observed". At first this is not obvious because TI, TKE are the same labels used on the x-axis of the respective TI, TKE row plots.*

We have updated figure 12 (figure 4 in our new version). We understand it is a lot of information to present, so we have made an effort to clearly explain it in the text and in the figure caption. Nevertheless, we feel it is an important figure because it summarizes our extreme case selection and the comparison of the model with the observations.

*figure 13: it is clear that the observed 1Hz equivalent loads (EQL) are lower, but strictly speaking that is only half the story. The other half is the frequency/distribution at which all the different events occur and together they result in the life time equivalent load. You could consider pointing that out as well.*

Yes, we fully agree. Using an EQL's conceal the underlying RFC and frequency information. This has been explained in the revision

*Regarding: 6. Comparison between aero-elastic loads at extreme events with loads from the reference spectrum:*

*I assume the extreme loads from DLC1.2 are based on at least 6 seeds per wind speed at 3 different yaw inflow angles (-8, 0 and 8 degrees)? In contrast, only 1x 10 minute realisation is used for the extreme events since they are directly based on a reconstructed/measured inflow field. I understand that in this context the authors want to demonstrate that extreme inflow conditions based on observations/high-fidelity simulations does not necessarily lead to higher loads when compared to the standards. I agree that is an important observation. However, I think it remains plausible that you could in theory create very similar inflow conditions as you have observed leading to extreme loads exceeding the DLC1.2 reference values. Since there is so much uncertainty and variation in turbulence levels across the rotor plane with smaller local "bursts" driving an outlier event (for example: https://dx.doi.org/10.1002/we.497). The more simulations you consider, the higher the likelihood you bump into such an event. From that perspective comparing extremes is only sensible when considering many 10 minute realisations. You could consider stating even stronger in the conclusions that many more aspects require more analysis, and that this study contributes to that bigger picture.*

Your assumptions are right: We used 6 seeds for the three different yaw angles. We will point that out in the revision. And yes, more realizations are needed to get a better picture of the extremes. We will also add this to the conclusions.

**Comments from reviewer 1**

For conciseness, we paraphrase the reviewer comments (*in italic)*:

General comments:

*Messy formatting.*

Our sincere apologizes, you are very right. Somehow Word is not doing what it is supposed to do but we have taken an effort to solve this as good as possible. Also apologizes for chapter 6 which occurs twice indeed. The chapter number for the conclusions should be 7 Chapter number

*Two different topics in one paper.*

See Appendix B for an elaborate view on this and how we implemented that in our revision.

*Points 1) 2) and 3): Agreement between calculations and measurement is not so good. (our paraphrasing of the comments)*

See Appendix A for an elaborate answer to this

*4) Acronyms.*

Apologies and yes we will spell them out in the revision. EQL is the equivalent load indeed. I remember this abbreviation was often used in the 1980's when this concept was introduced. Nowadays DEL may be more commonly used indeed.

*5) and 6) about 'special cases' and normal production cases*

Yes you are fully right, the ultimate loads generally do not come from the DLC1.2. What we are trying to say is that the ultimate loads from the extreme GRASP cases are lower than those from the

load set based on DLC1.2 only. But then they should definitely be lower than the ultimate loads from the full IEC load set because the full spectrum considers more cases and the ultimate loads can only become higher by adding load cases.

*7) Not a surprising result that feeding the turbulent fields of GRASP leads to lower loads because they have much lower turbulence than the standard class IA conditions.*

We respectfully do not fully agree with the conclusion that this is not a surprising result. Perhaps it is not surprising (this is of course up to a reader to decide for itself) but relevant, nonetheless.

We have taken the turbulent wind fields belonging to the most extreme cases from a year of LES results. We have also shown that our LES is able to capture the overall wind statistics at the site well. So, it is reasonable to assume that we produce realistic turbulent wind fields. We computed the loads based on these realistic turbulent wind fields and observe that they are lower than the standard IA class conditions would have given.

A possible explanation is, indeed like the reviewer suggests, that our cases come with TI levels that are lower than the standards. But this is just what the GRASP model (validated by the observations) tells us. The LLJ (and extreme shear) go hand in hand with low turbulence intensity. We fully agree that this is not a new insight but it is less trivial that the complete turbulent structure as modeled by LES, in a particular event like a LLJ with a non-trivial wind speed profile, the loads are reduced. As mentioned above, LLJ's are often considered negative in terms of loads.  We have added another picture that shows that both GRASP and the observations have similar TI - LLJ statistics.

*8) Elaborate on the dynamic controller*

You are right that details on the control are not given. The controller is believed to be representative since it has been designed with standard control design tools available at the partners in the INNWIND.EU and AVATAR projects with which they often design controllers for industry. Below rated wind speed, the turbine controller aims for maximum power production with variable rotor speed operation using a speed dependent generator torque setpoint (for optimum tip speed ratio) and constant optimal blade pitch angle. Above rated wind speed, the rotor speed and generator power are regulated to their nominal rating using constant generator torque and collective blade pitch control. We have included this information in the revision.

*9) Possible other discrepancies.*
We use the AeroModule which is a single code with an easy switch between the two different aerodynamic models: BEM and FVW. This assures the rest of the input (geometrical, structural, aerodynamic blade data, turbine data, control algorithm) to be the same. So yes we are sure that the differences are caused by the aerodynamic models. They are consistent to conclusions from the AVATAR project and although differences are large indeed the 14% difference is not larger than the differences found in the  AVATAR project. In the final report from AVATAR it is written that:
*Comparisons between aero-elastic calculations based on BEM showed a 15% over-prediction in fatigue loads compared to those from FVW, probably due to an inaccurate prediction of time-varying induction effects.*
See also Appendix C

*10) Table was unclear*
The deterministic component is the (mainly) the shear driven component. In the revised version of the paper we call it the azimuthally binned averaged component, which in the case of a linear system is similar to the deterministic component.

*11) Unclear paragraph on harmonics*
Yes I found it difficult to explain the 1P and 2P behavior. We have reformulated it as follows which I hope is better understandable

*…. Some further explanation is offered by Fig. 15. This shows a comparison between the azimuthally binned averaged flatwise moments for the LLJ and DLC1.2. Azimuth angle zero indicates the 12 o clock position. The rotor rotates clockwise so azimuth angle 90 indicates the 3 o clock position when looking to the rotor. The variation from DLC 1.2 has a 1P variation with a relatively large amplitude. This is the behavior of the flatwise moment in an atmosphere with 'common' vertical wind shear. The wind speed (and so the loads) decreases when the blade rotates from the vertical upward 12 o'clock (zero azimuth) position to the vertical downward 6 o'clock (180 azimuth). The flatwise moment increases again when the blade rotates from 180 degrees towards 360 degrees.*

*The variation in flatwise moment from the low level jet is very different. It shows a 2P variation with a relatively small amplitude. This 2P variation can be explained with the LLJ wind speed profile from Fig. 2 which shows the wind speed to be low at 0 degrees azimuth (the 12 o' clock position, when the blade is pointing vertical upward) and at 180 degrees (the 6 o' clock position, when the blade is pointing vertical downward). The wind speed is maximum at (approximately) hub height which correspond to azimuth angles of 90 and 270 degrees (i.e. the 3 o' clock and 9 o' clock position when the blade is standing horizontally)*
*This velocity variation is reflected in the flatwise moment. It is low at 0 degrees, high at (roughly) 90 degrees and 270 degrees and low again at 180 degrees. This leads to a 2P variation but the load amplitude is relatively small. Hence although the 2P load variation happens twice as often as the 1P load variation from the DLC 1.2. the lower amplitude of the variations lead to a lower fatigue*

*12) Work in progress and no novel conclusions*

We refer to our comments above and in Appendix C.

**Appendix A: Agreement between calculations and measurements**

We agree that the correspondence of a some of the presented 10-minute average wind speed profiles may be considered poor considering typical yearly averaged error metrics (modeled vs. observed wind speed) of a mesoscale or reanalysis dataset. However, we would like to point out that this can be expected when focusing on extreme cases, which are, by definition, the outliers in a dataset. The reviewer comments have made us realise that, in fact, a one-to-one comparison of extremes is not the most relevant. What is more relevant is the question: does GRASP capture, in a climatological sense, the (extreme) wind characteristics? In our revision of the paper, it is exactly this approach that we have taken: focus on the validation on the distribution (occurrences and magnitudes) of extreme wind events rather than a point-by-point comparison of single extreme events.

However, the reviewer comments were already indicating that the paper is trying to convey two messages in one paper. So, although an elaborate validation of extreme winds is an important and necessary next step that needs to be performed and presented in detail, we feel that putting too much emphasis in this would create a severe imbalance in the contents of the paper. To find the right balance between the validation of GRASP winds and the actual aero-elastic load calculation, we intend to include a number of extra figures showing the distributions of wind and turbulence and

90th percentiles of strong veer and shear events and to remove the one to one comparisons of the profiles. Furthermore, we give more emphasis to further GRASP validation in our recommendations for future work.

**Appendix B: Too many messages in one paper.**

The paper addresses too many messages which confuses the reader.

Yes, we understand and that is why we had added section 2.

In section 2 it is mentioned that, apart from demonstrating the combined GRASP-PHATAS tool, the research aims to investigate the impact of extreme events on the load spectrum but also the accuracy of a standard BEM model for the calculation of these events. These are two different subjects indeed which are to some extent unrelated. However, in our case the second subject is a spin-off from the first subject in view of the fact that we performed our calculations with AeroModule which is a code based on two different aerodynamic models: A BEM based model and a higher fidelity FVW model. So, insights on the accuracy of BEM are automatically obtained. We then feel it is a pity not to share these insights and this is supported by the comment from the second reviewer who writes that the consideration of BEM with disc averaged induction needs attention.

To make a long story short: We prefer to keep most of the results in the paper but we have prepared the reader in section 2 better on the fact that results are discussed on two unrelated subjects by explicitly mentioning this.

Another step with which we have taken to improve readability is to divide section 6 in two subsections:

- o Assessment of loads from extreme events
- o Accuracy of calculating loads from extreme events.

This is in line with the recommendation from the first reviewer

In order to improve readability, we have skipped the result for a rigid rotor and the discussion of flexibility effects. In retrospective we believe that these results and the associated discussion are less relevant and they only add to the confusion.

**Appendix C Conclusions are preliminary and not new:**

Our main take away messages are:

- The extreme events and in particular the LLJ's which we have considered do not give loads which exceed the loads from a standard design load spectrum. We think this is new because it is often stated that LLJ's have a negative effects on loads, see e.g. Duncan, J. November 2018 . But it is a preliminary conclusion indeed. We agree that the methodology needs further validation and we need to consider more aspects (more seeds, more years etc). We tried to make that clear by saying *that more research is needed to warrant a conclusion, especially in the validation of the on-site turbulent wind fields*. So, we agree that this is a preliminary conclusion indeed but we are not sure how to make this more explicit.

- BEM overpredicts fatigue loads for shear driven cases. We are confident on this conclusion. Indications for this overprediction have also been found in other studies, we refer to the EU project AVATAR and also the recently finished Dutch national project Vortex Loads

(Boorsma, K., Wenz, F., Aman, M., Lindenburg, C., & Kloosterman). However both AVATAR and VortexLoads  consider artificial load cases where the present study is based on extreme cases which result from a physical LES model. So, we think this is a new conclusion.

---

## Editor Decision (ED1)

Wes-2020-01

General

- Overall, the paper is very interesting in terms of the wind simulation work and comparison of wind simulation to data. The loads simulation is quite interesting – in particular figure 10 and the related discussion. The figures gemerally are quite interesting and overall the work is a nice contribution. However, the overall paper structure is weak and undermines the value of the paper to the wind research community. Paper should be restructured around the scientific question at hand rather than procedural description
    o "goal section" should be integrated to the introduction and expanded in terms of motivation
    o Methods description that appears distributed through introduction, section 3, and sections 4.1, 4.2 and 4.3, and section 5 should be moved to a formal methods section with the overall study scientific approach discussed before introduction of the wind simulation results
    o Two results sections:
        ▪ Wind simulation and validation
        ▪ Aero-elastic simulation and code to code comparison
    o Conclusions and recommendations
- Abstract needs a rewrite
- Consider having an editor (besides me) review for grammar and flow of the paper
- The use of bullet lists is far too liberal and in many places takes away from the paper flow and doesn't make sense (i.e. in the conclusion)
- More detailed comments by section are provided below

Abstract

- The abstract needs to be rewritten. It should succinctly describe what is interesting about the current work in terms of novelty and scientific value. It is currently descriptive and "unconventional wind conditions" is very vague. What does this mean? What about BEM modelling accuracy did you learn? What were the key findings of the work?
- Do not state in the abstract the organizations involved, it is irrelevant and evident already in the author list

Introduction

- The paper lacks a proper motivation. Why is this interesting? Why should a reader care? What is the problem that this article is trying to solve?
- First sentence is awkward. Recommend for example: "This article investigates the loading of an offshore 10 MW wind turbine in response to extreme wind events in the North Sea."
    o Generally, the article could do with a clean-up for grammar, flow and readability

- Lack of explanation of the use of the LES model and its validation – this is likely in the cited paper but it would be good to explain why this data set is being used for this study and what about its use will lead to novel insights
- The entire discussion from line 27 to line 37 reads more like a methods section except lacking in detail.
- "conventional procedure along the IEC standards" is vague
- This is the objective of the paper and should be elevated and an introduction properly motivate the objective: "By comparing the loads in response to the extreme events with those from the conventional design load spectrum, the importance of extreme wind events can be assessed for practical (load) purposes."
- Git rid of the bullets and write the paper outline in paragraph form as is the convention

Goal

- Please consider removing the section and integrating with the introduction. Either that, or retitle the section though the first option is preferable.
- The first bullet is a code mechanics and not scientific contribution. Rephrase or eliminate.
- The second bullet is the main paper contribution and should be elevated. Everything in the paper should be structured around this. It should be moved to the introduction along with a contextual discussion of why site-specific analysis like this would be necessary versus following the IEC standard approach. Also why its not already done in practice should be discussed as well and then how this paper provides a first step towards enabling site-specific analysis for x, y, z purposes.
- The third bullet is important as well – can we rely on BEM for accurately assessing loading like we would see in such conditions as explored here? Elaborate on why the code comparison is being done.
- Delete: "Hence it should be realized that, apart from demonstrating the combined GRASP-PHATAS tool," – this is not a scientific contribution
- Rewrite the last paragraph of this section – the two are absolutely related if the overarching objective is the investigation of loads assessment in realistic site-specific offshore environmental conditions

Reference turbine and location

- Too many scientifically irrelevant details about the AVATAR project in the first paragraph – if necessary to include, do so via a footnote
- The low induction concept is not well explained. Is this relevant for the current study? In what way? While many readers likely know what this is, it is not ubiquitously understood and merits further explanation
- Many of the details of the design would be better stated in a table form rather than sentence form – it will save space and be more digestible
- The mast and site location are introduced without proper motivation – was the mast data used for validation at some point? Is the mast of particular importance to this paper? If so how? All of this could be better described if integrated into a methods section (see general comments)

Calculation of loads for extreme events

- Line 126-127 – how were these events identified? Was all this work with GRASP and LES part of this study? If so, it should be properly discussed in terms of the validation – a single plot (figure 2) seems quite limited and there is no discussion of it. If not, please reference other literature where this has been done more thoroughly.
- While using the site data is quite interesting, it is unclear if the year chosen is quite representative as extreme events might occur in other years – why was this year selected? Is it representative of the extreme conditions the turbine will face? What are the limitations of the approach? (it is very cool work, but the discussion is currently lacking)
- How were the "extreme" cases selected? Can you discuss a bit more how you came to select these? How do they compare with what the standard would recommend?
- "For the selected LLJ case the corresponding observed wind profile does not show a jet-like profile." Can you discuss or explain?
- Figure 4 is really great. I would like to see a bit more discussion:
    o Can you explain at all the mismatch between observations and simulations for the different cases?
    o When indicating that the conditions show presence of a LLJ, please provide more detail as to how – it will not be apparent to many readers.
    o What are the closest DLCs in the standard for each of these cases or where would you expect to see them surface in IEC standard analysis? There are a couple of comments linking the cases to the standard but it could be further developed
    o "In contrast, for the LLJ case the observed values of TI do increase with height, which would be much harder to explain. " – please try…
- ERA5 analysis is presented but not really discussed at all – either exclude it or refer to it in discussions
- Line 208: Can you define climatology – i.e. change to: "In this section, we move from comparison of isolated 10-minute records to the climatology, (climatology definition), of extreme wind events from the GRASP LES versus the MMIJ observations." Generally it refers to the study of changing climate over time, but I don't think that is how you are using it here.
- Line 214: Can you explain why GRASP fairs better in capturing strong veer than ERA5?
- Line 236: ERA5 underestimates speed of LLJ – why? And what does GRASP do differently that enables it to better capture these events?
- Line 250 – somewhat irrelevant details about code history- footnote it – better would be to describe model approach for aero-elastic analysis – modal? Multi-body?
- There are a lot of improvements to BEM beyond the works cited in lines 259. As BEM / vortex comparison is a key part of the study, I suggest an elaboration on this and its inclusion in the methods section rather than here sandwiched between wind simulation analysis results and aeroelastic analysis results
- The whole section of 4.2 is methods versus results – it is recommended to move to methods section
- Will the AWSM model serve as "truth" here?
- No need to have a separate section 4.3, it can be part of the prior discussion of the overall approach to the aeroelastic simulations and also should be moved to a methods section

Calculation of reference design load spectrum

- Move this entire section into the methods section as a subsection
- "almost complete set of design cases" is vague
- It is unclear that you are redoing this analysis or just reusing the results of the prior referenced study (i.e. TurbSim is mentioned in 4.3 for Whiffle input and SWIFT in 5 for IEC).

Comparison between aero-elastic loads at extreme events with loads from the reference spectrum

- This section could be significantly improved and restructured by focusing first on the whiffle based simulation results and comparison of BEM and AWSM. Then move on to the standard and comparison.
- Isn't it more common to use the terminology flapwise than flatwise?
- For figures 8 and 9, consider removing the line for the reference andjust having the dots
- Im not sure why bullets are used in lines 317 to 331 as the comments do not seem to have a common point of relation. They are simply continued discussion of the results and would be better in paragraph form.
- The entire comparison in figures 8 and 9 seems a bit odd – its an apples to oranges situation.  It would be good perhaps to separate the figures out and show the loads from the whiffle input data first and then from the standards analysis second. It would be better to see in the first case, the differences between BEM and vortex for the cases and a discussion as to why differences surface.
- The overall DEL comparison is a struggle
    o How are the DELs calculated in each case and is this really apples to apples? A fair comparison?
    o The absolute values in the tables are hard to interpret except for the fact that for DLC1.2 they are much higher than for LLJ and then for BEM to AWSM respectively… is there a better way to present this data?
    o Based on this study, do you think the study is really overly considered or if more site data were taken (over a much longer time period) would we see loading closer to DLC 1.2?
- In section 6.1 the use of bullets is again a bit odd. Also, some of the discussion seems to be tied to the prior table and figures and some to figure 10, why is this a separate sub-section?
- Figure 10 and discussion is interesting and well-discussed. Makes more sense than the discussion related to figures 8 and 9 and subsequent table
- Redundant section 6.1 title for accuracy. Again, does this need to be a subsection? I would move this discussion further up and discuss it where the data for BEM versus AWSM is shown in figures
- For lines 398-401, is there evidence of this in the results that you have? Are these commonly believed explanations ones from the cited papers?

Conclusions and recommendations

- Get rid of the bullets

- Emphasize the scientific learnings. Anything code related is a support element to enable the analysis
- Lines 434 to lines 439 – this is a much better description of what this paper is about than anything written prior to. Pull some of this into the introduction and abstract
- Line 438-439 – what is the value of doing the simulations based on physical wind models? This is a key aspect of the paper but it is not properly motivated anywhere
- Future work should be the last paragraph
- Author contributions and acknowledgements should be footnote or its own section

---

## Author Response (AR2)

We again thank the reviewers for their positive feedback and for their final comments on how to improve the manuscript. The comments are greatly appreciated. With this document we explain how the comments are included.
For convenience we have submitted the document with track changes so that that the changes can be seen clearly.

**Comments from reviewer 1:**

Line 61: "In this way it can be assessed whether these extreme events yield loads which exceed the design load spectrum. If true, this might lead to a recommendation on adaptation of the load spectrum with events found in the present study." And if not, as it turns out? Adjusting the DLCs? Reduce safety factors? One of my concerns from the first review was the weak conclusions of the work. This aspect has greatly improved in the revised paper (thank you), but in my opinion the key takeaways may still benefit a few more thoughts. Given the results that you show, what do you recommend for the standard? What should a designer do after reading your paper? You list a more accurate wind modeling and the coupling to higher fidelity solvers among the next steps. But what do we do for wind parks where turbines are not isolated, but experience waked inflows? How would you model these phenomena in your numerical framework?

> *See the additional text at line 61 and the additional text in the conclusions*

Line 114: "ERA5 boundary conditions". What are they exactly? Also, no conclusion talks about them. Should one bullet point discuss the differences between GRASP and ERA5 outputs?

> *See the additional text*

Figure 2 and Figure 6 have the top of the legend cut (1000 instead of 10000). The whole paper would also look so much nicer if all figures were generated with the same style, Matlab or Python or Excel or Paraview or ... It could be a very easy and so useful upgrade.

> *A very good suggestion: All figures are now made with the same tool*

Line 181: "However even a turbulence class C (the lowest possible turbulence class in IEC) leads to turbulence intensities which are still far below the extreme ". Shouldn't the word be "above"? The extreme values observed are below the values from turbulence class C (and A), not vice versa, no?

> *A very clumsy and confusing mistake. Apologizes! Has been corrected*

Line 279: were the original loads run in Phatas? I tried to access the AVATAR deliverables to check myself, but the website seems to be down (or not accessible to me?). If they were, I find the fact that you rerun the loads irrelevant to the reader. I understand it was a useful step for you to avoid inconsistencies, but the reader does not gain anything here. I would remove the sentence at line 281/282.

> *The original loads were run with Phatas indeed but I agree that the value is internal and it has no value for outside readers so I removed the sentnec*

Line 289: "The yaw angle is prescribed to be 8 degrees in line with the IEC standards." Isn't this condition for misaligned load cases only? That should not be true for DLC 1.2. What am I missing?

*As a matter of fact the IEC standard, section 7.4.1, second alinea says the following: ""In addition, deviations from theoretical optimum operating situations such as yaw misalignment*

*and control system tracking errors shall be taken into account in the analyses of operational loads. We at TNO generally use 8 degrees and this 8 degrees is based on the GL standard section 4.5.1 which states: "Yaw misalignment and hysteresis shall be considered in the yaw movement. If values for the turbine type cannot be specified, yaw misalignment of - 8°, 0° and +8° evenly distributed in ± 8° shall be applied (DLC 1.1 to 1.3 and 1.5 to 1.7)"*

*Our sentence was a bit cryptic indeed but I feel there is not much value in a detailed explanation for the purpose of the present study. I replaced the sentence by: A small yaw angle of 8 degrees is included to account for yaw control tracking errors.*

Figure 9: typo in the label of the y axis "Exterme".

Corrected. Thanks!

Line 361: "The variation in flatwise moment from the low-level jet is very different." Azimuthal variation? Or variation compared to what?

*This is the azimuthal variation indeed. And it is compared to the variation which results from DLC1. We added an explanation*

**Comments from reviewer 2:**

* line 114: to what is \textit{Dutch North Sea area with ERA5 boundary conditions} referring to? Either a reference or introduction seems lacking. Could be this is obvious of readers with a stronger meteorological background, but it is not obvious to me.

*Reference is included*

* For the scatter density plots: I assume the color refers to the number of occurrences? Could you add this to either the caption or add a label to the color bar in the figure?

*Has been added*

*line 316: is it possible you accidentally forgot to remove this reference to a sequel report?

*Actually This is not a sequel report but it is a completed report. We think it has relevance to let the reader know that we have simulated a reduced reference load set where a full IEC reference load set would increase the extreme loads even more.*

*Finally: The turbulence as fed into AeroModule is frozen indeed like it is in the turbulence from a conventional wind simulator. We have added a remark on that in the conclusions*

---

## Author Response (AR4)

**Reply to editor**

We are thankful for the careful and precise review of our work and the numerous suggestions for improvements. We completely agree with what we believed was the the overall message of your comments: the work is good, but the presentation needs to be improved on the aspects of language, organisation, structure and positioning with respect to the scientific literature (our paraphrasing). We have therefore thoroughly rewritten and restructured the paper in the light of your comments.

We believe we have addressed almost all of both your general and your specific remarks. Here and there, we have taken the liberty to keep our original formulations, because after careful considerations we decided that we preferred those over suggested improvements. An example of this is the choice to keep suggestions for future research throughout the conclusions section and not bundle them into a separate paragraph. The reasons is that we feel that this way of presenting makes them more 'naturally' connected to specific finding the recommendations originates from.